# Geophysical Research of Secondary Deformations in the Post Mining Area of the Glaciotectonic Muskau Arch Geopark—Preliminary Results

Jan Blachowski [1,*], Ewa Warchala [2], Jacek Koźma [3], Anna Buczyńska [1], Natalia Bugajska [1], Miłosz Becker [2], Dominik Janicki [2], Paulina Kujawa [1], Leszek Kwaśny [2], Jaroslaw Wajs [1], Paweł Targosz [4] and Marek Wojdyła [4]

[1] Department of Geodesy and Geoinformatics, Faculty of Geoengineering, Mining and Geology, Wroclaw University of Science and Technology, Wybrzeże Wyspiańskiego 27, 50-370 Wroclaw, Poland; anna.buczynska@pwr.edu.pl (A.B.); natalia.bugajska@pwr.edu.pl (N.B.); paulina.kujawa@pwr.edu.pl (P.K.); jaroslaw.wajs@pwr.edu.pl (J.W.)
[2] KGHM CUPRUM—Research and Development Centre, Gen. W. Sikorskiego Street 2-8, 53-659 Wroclaw, Poland; ewa.warchala@kghmcuprum.com (E.W.); milosz.becker@kghmcuprum.com (M.B.); dominik.janicki@kghmcuprum.com (D.J.); leszek.kwasny@kghmcuprum.com (L.K.)
[3] Polish Geological Institute—National Research Institute, Lower Silesian Branch, al. Jaworowa 19, 53-122 Wroclaw, Poland; jacek.kozma@pgi.gov.pl
[4] Geopartner Geofizyka, Skośna 39B, 30-383 Krakow, Poland; pawel.targosz@geopartner.pl (P.T.); marek.wojdyla@geopartner.pl (M.W.)
* Correspondence: jan.blachowski@pwr.edu.pl

**Abstract:** Underground and opencast mining adversely affects the surrounding environment. This process may continue even decades after the end of actual mineral extraction. One of the most significant effects of ceased mining are secondary deformations. Safe, new development of post-mining areas requires reliable information on potential deformation risk zones, which may be difficult to obtain due to a lack of necessary data. This study aimed to investigate and understand the secondary deformation processes in the underground mining area of the former "Babina" lignite mine, located in the unique glaciotectonic environment of the Muskau Arch, in western Poland. A combination of GIS-based historical mapping, geophysical 2D/3D microgravimetry, and Electrical Resistivity Tomography (ERT) measurements allowed the identification of subsidence-prone areas and the determination of potential factors of sinkhole development. The latter are associated with anthropogenic transformation of rock mass and hydrogeological conditions, by shallow underground mining. The results confirmed that multi-level mining of coal deposits in complex and complicated glaciotectonic conditions cause discontinuous deformations, and may be hazardous as long as 50 years after the end of mining operations.

**Keywords:** microgravimetry; ERT; brown coal; post-mining; deformation; geopark; Muskau Arch

## 1. Introduction

Post-mining areas, resulting from both underground and open-cast operations, experience negative consequences even decades after the end of mining operations [1–5]. These effects include, among other things: flooding of the subsided areas [6,7]; sinkholes and other forms of discontinuous deformations, as a result of sudden destruction of underground workings, especially shallow ones [8,9]; secondary deformations (uplift), as a result of the restoration of groundwater levels [10–13]; erosion of mining waste heaps; fires related to coal mining waste heaps; pollution caused by waste heaps [14–17]; difficulties with the introduction of vegetation and degradation of vegetation conditions due to unfavorable soil characteristics; and subsidence and flooding [18–21].

These processes pose a threat for new development and use of post-mining areas. As precise information on the location and extent of former excavations is often incomplete or

lost, it is difficult to properly plan measures to counteract and protect against the potential, and often sudden, secondary changes of post-mining areas. In addition, as these areas are no longer monitored, e.g., by mine surveying services, the extent and intensity of these processes remain largely unknown.

Secondary subsidence is one of the most threatening processes in post-mining areas. It is deformation of the land surface caused by the readjustment of rocks above voids left by underground mining activity, and is usually sudden. The collapse of a mine working could be the result of factors such as: changes in hydrogeological conditions related to water infiltration into the rock mass; a loss of support for old, shallow excavation workings; weakening of the rock mass properties, as a result of weathering and rheological processes; development of voids, as a result of displacement of backfill material in improperly liquidated shafts; dynamic loading of the rock mass with communication vibrations; or mining tremors [22–25].

Studies of post-mining areas with a focus on establishing of the extent of former underground mining activity, mapping present-day deformations, modelling and predicting the potential for secondary deformations, and risk assessment of post-mining areas have been conducted, with different measurement, processing, and analytical methods. Land deformation measurements in post-mining areas are carried out with terrestrial (e.g., levelling, geophysical), as well as satellite and/or airborne techniques [8,25–34].

Geophysical surveys are an important aspect of studies of post-mining areas, as they enable the continuous and non-invasive measurement of specific parameters of the rock mass. These parameters, depending on the chosen technique, are usually determined from the processed data. Ground-based surveys may take one of many forms [35,36], e.g., seismic surveys based on measurement of the propagation of seismic waves produced by human-generated impact or vibration in the ground; gravity surveys using a gravimeter to measure the gravity field to determine variations in rock density; electromagnetic surveys, which involve examination of changes in ground conductivity induced by electromagnetic fields; Ground Penetrating Radar (GPR) measurements; or resistivity methods, based on measurements of the variation of electrical properties of the ground. These methods have been applied for many years to define lithological boundaries and tectonic features, as well as to locate post-mining voids and underground infrastructure in the rock mass, such as drifts or shafts, and the results aid in the assessment of risk of ground subsidence. A review of geophysical methods for the detection and mapping of underground mine workings has been presented by [37]. Noteworthy examples of the application of these methods in studies of post-mining areas include:

- Location of abandoned transport and ventilation shafts in a post-mining lead-zinc area in the USA, with seismic reflection and GPR methods [38];
- Location of underground coal mine workings in India to assess the risk of crown hole subsidence occurrence, with multi-electrode resistivity imaging (RI) and Ground Penetrating Radar (GPR) [39];
- Characterization of derelict coal mine workings in the UK, that included locating an abandoned coal mine shaft [40];
- Detection of post-mining voids using the microgravity technique in karstic environment [41];
- Long-term microgravity and geotechnical monitoring of relict salt mines in the UK [42];
- Estimation of sinkhole occurrence risk in post-mining areas of shallow hard coal exploitation in Poland, with gravimetric and seismic measurements [43];
- Assessment of mine collapse (sinkhole) risk due to voids in underground shafts and the surrounding rock mass [24];
- Detection of abandoned strontianite mine workings in Germany, with a combination of seismic refraction tomography and resistivity tomography [44];
- Identification of voids from underground coal mining which pose a threat of sinkhole occurrence on the surface in Upper Silesia (Poland), with GPR methods [45];

- Mapping of subsidence and sinkholes related to post-mining voids resulting from underground coal mining in South Africa, with High-Resolution Seismic Reflection surveys [46];
- Mapping of abandoned mine underground workings and unconsolidated zones in abandoned coal mining areas in Bulgaria, with application of electrical resistivity tomography (ERT) [47];
- Analysis of the occurrence of voids or loosening zones in the rock mass near an abandoned mining shaft [48];
- Combination of geodetic, GNSS, and geophysical measurements for complex monitoring of mining areas [49].

Geophysical methods have advantages and disadvantages. The main advantage is the possibility of "seeing without drilling". All of the mentioned geophysical methods provide the distribution of petrophysical parameters below the surface in the form of cross-sections and maps. For studies of post-mining areas, the gravity and seismic methods are the most valuable, because density and/or velocity models can be obtained. Changes in the density of rocks are directly related to the remains of underground works. Resistivity methods are useful for hydrogeology, as electrical parameters depend on the conductivity of the subsurface. Considering the disadvantages of geophysical methods, the problem of resolution should be mentioned here. The more a priori information (for instance, from existing boreholes), the better the obtainable results from the modelled data. However, for so-called "shallow geophysics", very trustable information can be provided from surface geophysics, without boreholes. As proved in the mentioned publications, a combination of geophysical methods can be used for the development of models describing the underground situation in post-mining areas.

The research presented in this paper aimed at creating a digital inventory of the selected post-mining workings, located in a unique postglacial environment, and mapping the extent of the rock mass disturbed by underground brown coal mining, with the objective of identifying weaker areas susceptible to present-day mine collapse and secondary land deformation. For these purposes, a 3D geospatial database of underground workings, based on available historic mining documentations, was created in GIS, and 2D/3D microgravimetry, followed by electrical resistivity tomography (ERT) geophysical measurements, was performed, and their results were analyzed and interpreted. The study area, methodology, and results, with their discussion, are described in Sections 2–4, respectively.

## 2. Study Area

The study area, which is the former brown coal mine "Friendship of Nations—Babina", is located in western Poland, close to the border with Germany (Figure 1). The mine was in operation from 1921 until 1973. It is a unique post-mining area, as, initially, the brown coal was mined through a system of shallow underground workings that followed the complex geometry of coal deposits, a result of glaciotectonic processes, and were accessed through a system of inclined shafts. The depth of the underground workings reached up to 100 m below the terrain level. Since the 1960s, the deposits were also mined from the surface in several open pits. The overlying waste rocks were deposited in external and internal waste heaps. After the end of mining, due to economic reasons, the area was subjected to reclamation measures aimed at the restoration of vegetation. These activities included, in the first phase, the levelling of steep slopes in the pits and on the heaps. In the second phase, the pits were filled with water and trees were planted in the remaining area. The second stage was never finished. Details of the reclamation activities are described in [50]. Four study fields were established in the scope of the research project within the border of the former mining field, and the presented study concerns geophysical surveys carried out in test field three (Figure 1). This test field was established in the area of underground mining activity.

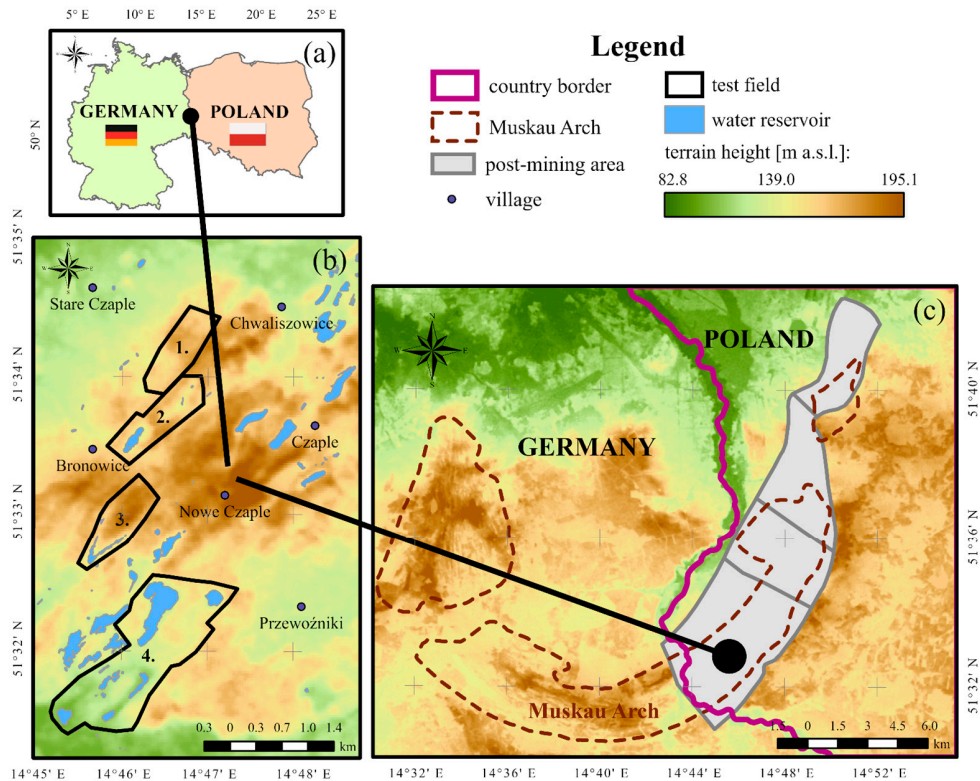

**Figure 1.** General location of the study area: (**a**) location of the Muskau Arch (**b**) test field three; (**c**) extent of the Muskau Arch.

In terms of geology and geomorphology, the study area is located in the south-east part of the glaciotectonic structure called the Muskau (Muzakow) Arch. It is a terminal moraine in the shape of a regular arc, open to the north, with an arm span of about 20 km, length of approximately 40 km, and width of around 5–6 km. The central part the arch is divided by the valley of the Nysa Luzycka river, which is also the border between Poland and Germany. The structure was formed as a result of the load exerted by a moving glacial ice sheet during the Marine Oxygen Isotope Stage (MIS) 12, also known as the Elsterian glaciation [51]. The original moraine, that reflected the shape of an ice lobe, was eroded and modelled during the following MIS 6 (Salle) glaciations. In the effects of glacial erosion processes, the height of the moraine was greatly reduced, and deeper glaciotectonic structures, dating back to the MIS 12 period, were exposed in numerous places. The ablation moraines, often perpendicular to the older glacial structures, are a trace of the glacial accumulation from this period.

The Middle and Upper Miocene deposits, deformed during the MIS 12 and MIS 6 glaciations, were tilted and raised from a depth of about 150 m below ground level. The deformation forms of the Paleogene and Neogene deposits are of different sizes, ranging from over a dozen meters to several kilometers. These are mainly continuous compression structures, such as folds and diapirs, as well as discontinuous structures that are dominant in the area, such as overhangs and glaciotectonic scales (Figure 2). These glaciotectonic structures are clearly visible in the present-day relief of the terrain. Distinct geomorphological forms include elongated lowering of the terrain in the zones of disturbed brown coal deposits. Their origin is related to the process of compaction of brown coal caused by weathering. The width of the lowered zones reaches 30 m, with a length of up to 2 km and a depth between 3 and 5 m. They are filled with Holocene alluvial deposits and peats. These subsided forms are divided by deposits of ablation moraines. Together, these represent a stripe-like arrangement of the geological relief and landscape composition [52,53].

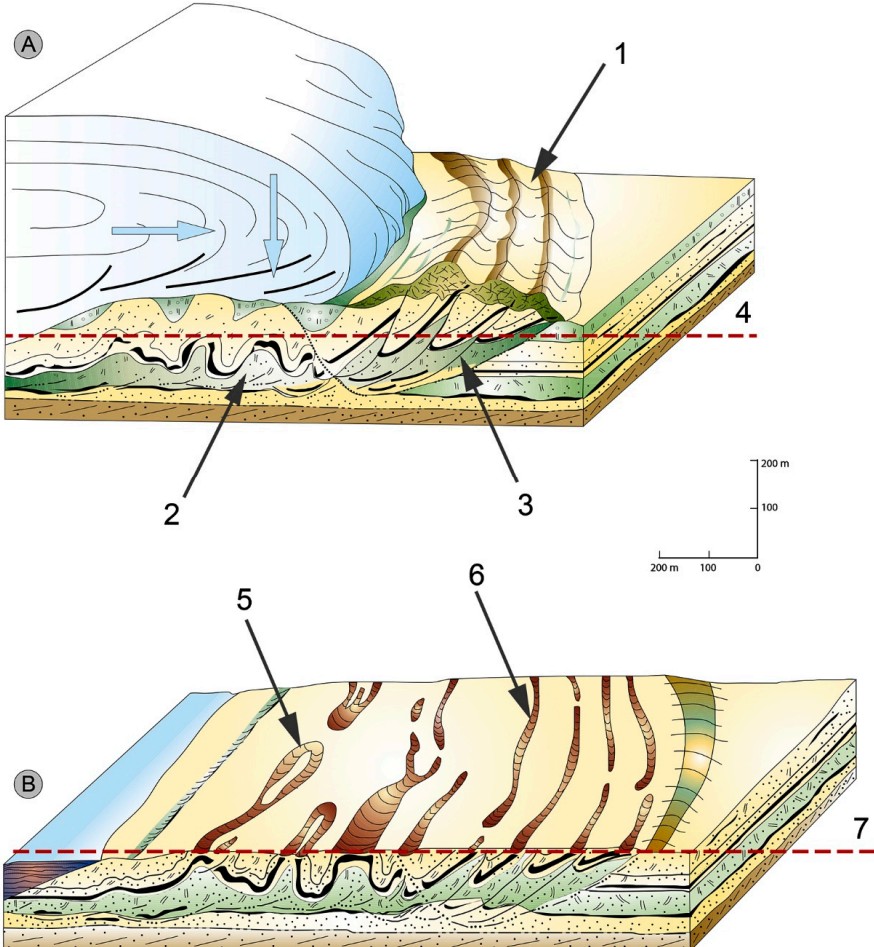

**Figure 2.** General ideological model of the deformation and erosional processes in the area of the transboundary Muskau Arch ([51] after [53]): (**A**) The approximately 430–510 m thick glacier dislocates the horizontal layered bed of tertiary and quaternary deposits by loading and horizontal push. The glacier acts down to a depth of about 270–290 m beneath the ice base. 1—terminal moraine; 2—fold and diapir structures; 3—scale structures; 4—level of moraine erosion. (**B**) The contemporary shape of the coal seam outcrop, depending on the type of glaciotectonic structures. 5—brown coal deposits associated with folds and diapirs; 6—brown coal deposits associated with scales; 7—present day level of terminal moraine erosion.

The specific study area is test field three, which is located in a glaciotectonic complex called the Nowe Czaple zone, and is dominated by fold-type structures. In the morphology of this area, natural and anthropogenic zones of lowered ground surface are clearly visible. These set the V shaped course of brown coal deposit outcrops, which is most probably a large fold structure consisting of two (northern and southern) arms, with the area between them elevated and made up of quaternary sand and gravel formations. Within the two arms of the V shaped fold structure, multi-level underground mining of brown coal deposits was conducted in the period of 1963–1972. In the present-day topography of the area, above the complex architecture of underground drifts and shafts, elongated subsidence basins, partly covered with water, are clearly visible [53]. The area of test field three is shown in Figure 3.

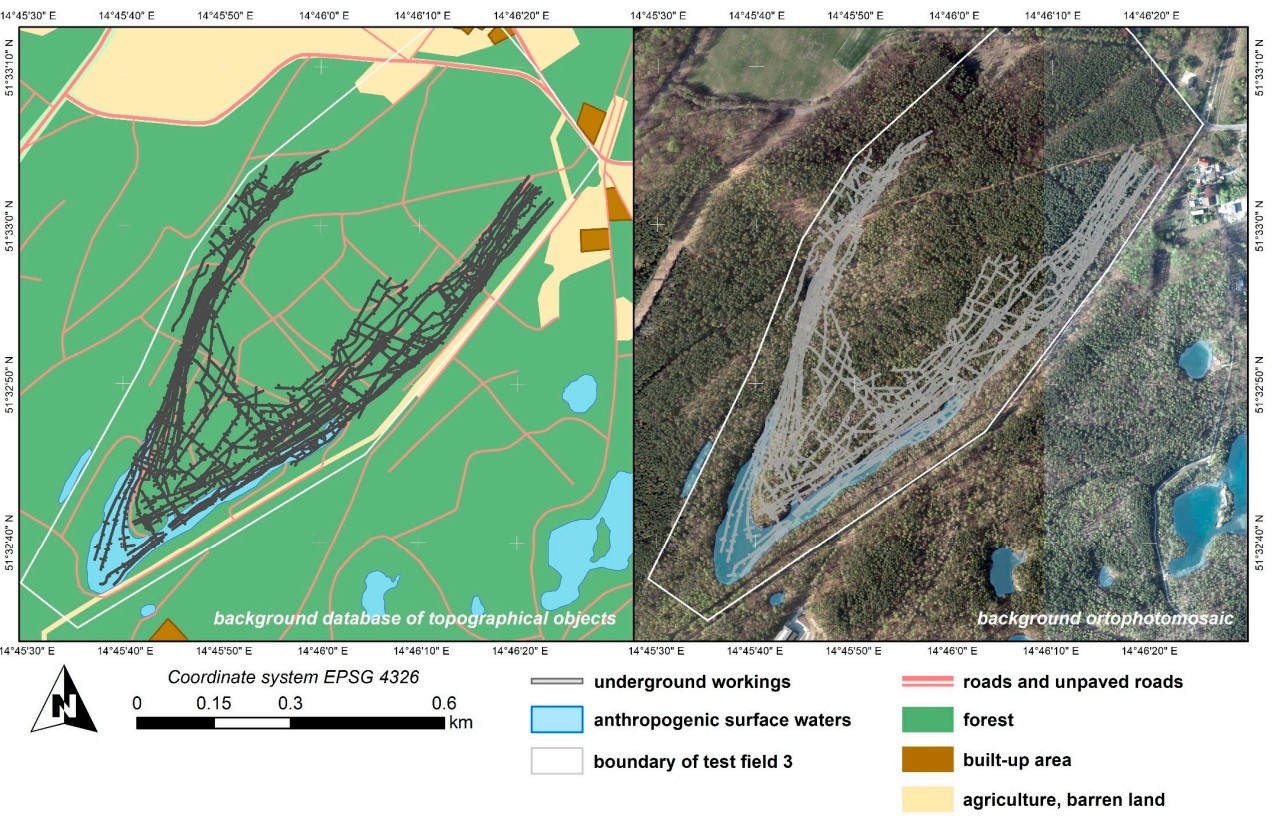

**Figure 3.** Area of test field three in relation to topography and documented underground workings; left, topographical background; right, orthophoto mosaic background.

## 3. Materials and Methods

The research methodology adopted for the study of the condition of ground disturbed by underground mining in the selected test field consisted of the following stages:

1.  Development of 3D geodatabase of underground workings;
2.  Setting out geophysical surveying lines and sites;
3.  Geophysical 2D microgravimetry surveying of profile lines;
4.  Geophysical 3D microgravimetry surveying of test sites;
5.  Geophysical surveying with electrical resistivity tomography;
6.  Analysis and interpretation of the condition of ground disturbed by underground mining.

### 3.1. Development of the Geodatabase

The geodatabase was prepared with ESRI ArcGIS Pro software, licensed to the Wroclaw University of Science and Technology, based on archival cartographic documentation comprising of 67 scanned paper maps representing underground and open-pit workings. The scanned maps were obtained from the State Mining Authority archives, and are drawn at a scale of 1:1000. The preserved and available maps do not cover the entire area of the study (all test fields). However, for the analyzed test field (three), the documentation is complete. The scanned and georeferenced maps were used to create vector-type features representing location and 3D geometry of the former underground workings. An example map is shown in Figure 4, and a visualization of the underground workings in this test field is shown in Figure 3. In addition, the location of geological profiles made prior to the start of the mining operation in the area, obtained from the Polish Geological Institute–National Research Institute archives, was included in the geodatabase. Other data used for this study included DEM with a 1 m resolution from the year 2011, as well as Database of Topographical Objects at a scale 1:10,000, both downloaded from the Head Office of Geodesy and Cartography geoportal.

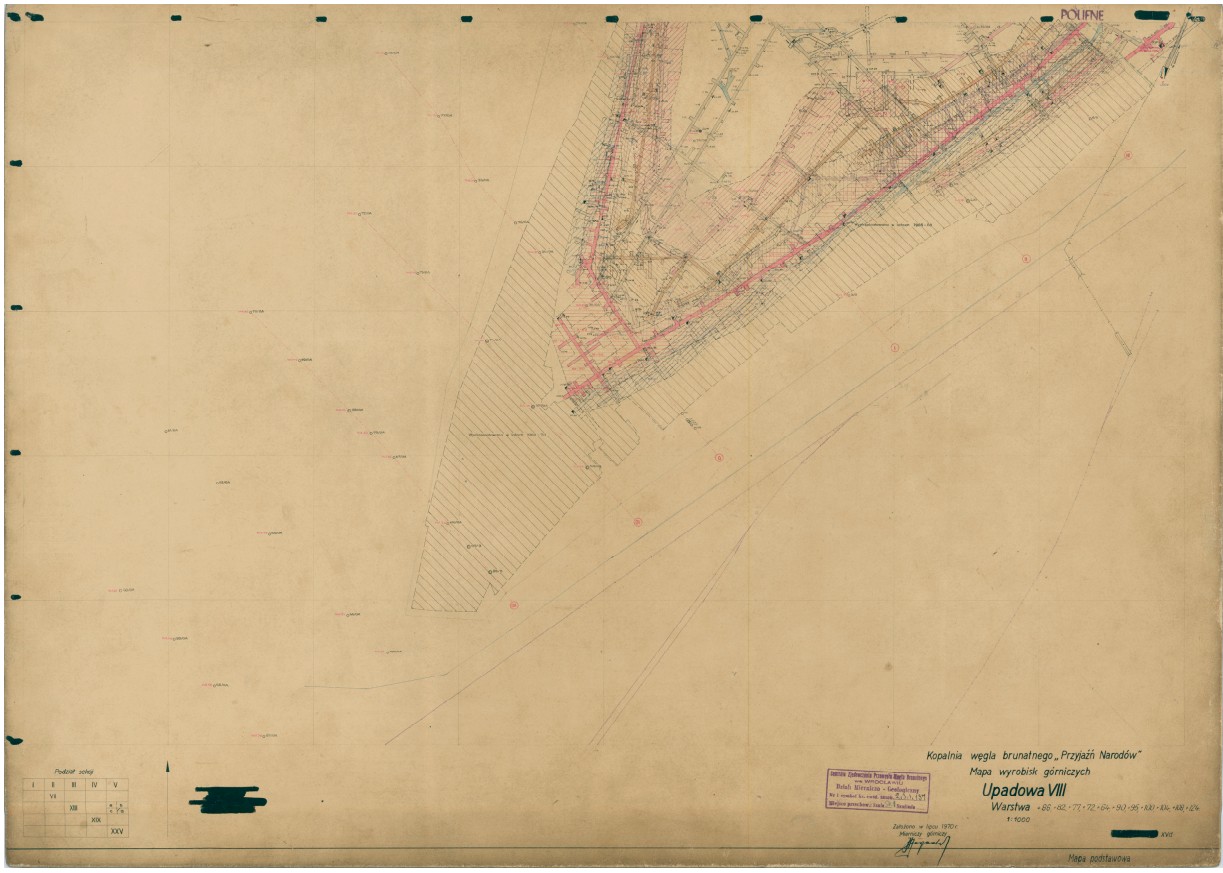

**Figure 4.** An example of source documentation; a section of mining map depicting underground workings.

### 3.2. 2D and 3D Geophysical Microgravimetry Surveys

Microgravimetry is a method used for the search and recognition of shallow underground forms of small size, often of anthropogenic origin. The results of microgravimetric studies are used to detect voids and discontinuous deformations. Conclusions concerning geological structure of the studied area are developed in the form of maps and density models showing subsurface distribution of rock masses. Details of the method can be found in [37,41,42,54], and example applications of the method in studies of post-mining areas are stated in the Introduction section.

Based on the analysis of the location of former underground operations, in relation to present-day topography and archival geological profiles, a geophysical survey network for 2D and 3D microgravimetry surveys was designed. This included profile lines that were set out at an interval of approximately 100 m, and approximately 10 m spacing in a given line in the area of test field three. Furthermore, three detailed test sites were set out for 3D gravimetric surveys in a quasi-regular 6 × 6 m grid, including one site where a sinkhole was spotted in late 2019. The location of the profile lines and test sites is shown in Figure 5.

The gravimetric field surveys consisted of the establishment of 4917 measurement points and GNSS measurements of their location; 3900 points were located along the 2D profiles, and 1017 in the detailed test sites. For the microgravimetry surveys, two Scintrex CG-5 Autograv gravimeters were used. The gravimetric measurements, at scattered points, were made with the method of intermediate points. Thus, each measurement sequence began and ended with measurement on a base point with a known value of the acceleration of gravity g. To determine the accuracy of measurements, acceleration of gravity g values was determined twice at selected positions, distributed evenly over the entire area of study. The mean square error of a single measurement amounted to ±0.006 mGal. For data processing, Schlumberger Petrel (Grav&Mag plug in) and ZOND software were used.

The results of interpretation of the obtained geophysical data are presented in the form gravimetric maps in the Results and Discussion section. The prepared gravimetric maps highlighted the geostructural forms, and helped to define the range of possible land deformations that were further studied with the Electrical Resistivity Tomography (ERT) surveys.

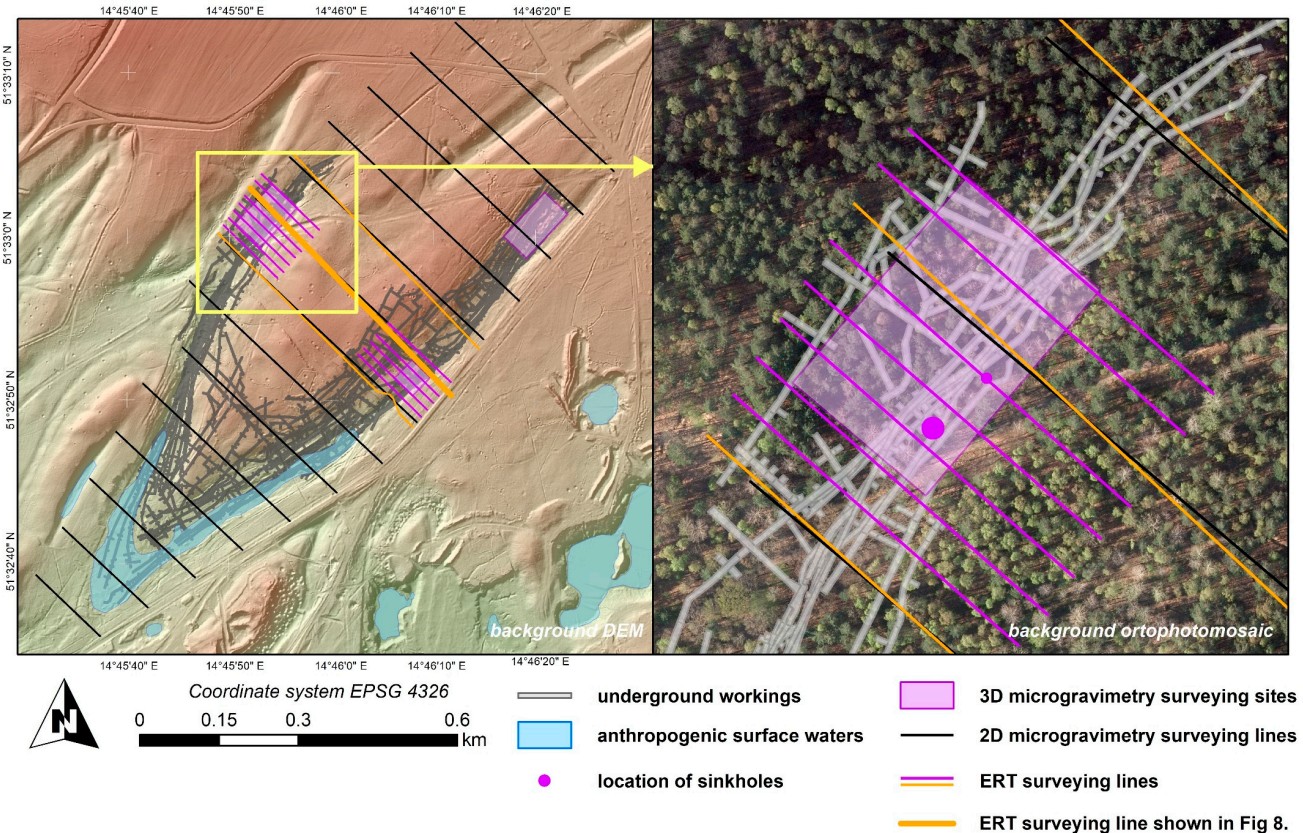

**Figure 5.** Location of geophysical surveys in relation to topography and documented underground workings in test field three.

### 3.3. Electrical Resistivity Tomography Surveys

The purpose of electrical surveys is to determine the subsurface resistivity distribution by making measurements on the ground surface. From these measurements, the true resistivity of the subsurface can be estimated. The ground resistivity is related to various geological parameters, such as mineral and fluid content, porosity, and degree of rock water saturation.

Resistivity measurements, or so-called resistivity sounding, are normally made by injecting a current into the ground through two current electrodes and measuring the resulting voltage difference at two potential electrodes. From the current $I$ and voltage $V$ values, an apparent resistivity $\rho_a [\Omega m]$ is calculated:

$$\rho_a = k \left( \frac{V}{I} \right), \tag{1}$$

where $k$ is the geometric factor, which depends on the arrangement of the electrodes. Using measurements of $I$ and $V$ for different $k$, we can estimate the resistivity changes in vertical direction. A more accurate model of the subsurface is a two-dimensional (2D) model, where the resistivity changes in the vertical direction as well as in the horizontal direction, along the survey line. Details of this method can be also found in [55–57], and example applications of the method in studies of post-mining areas are stated in the Introduction section.

In our case, ERT measurements were conducted in 15 profiles with a total length of 4413 m. The location of profile is shown in Figure 5. In the surveys, an asymmetric three-electrode scheme was used, with the spacing of the electrodes along the profile equal to 3 or 5 m. The scheme is presented in Figure 6. Data were recorded with the ARES II 850 W Automatic Resistivity System. The adopted methodology aimed for semi-detailed reconnaissance of the rock mass up to a depth of 70–80 m below surface level, in the central part of test field three, and detailed identification of test sites to a depth of 30–40 m below surface level. The collected data were processed by 2D inversion in profile lines. As a result, resistivity models of the ground were obtained. Electrical resistivity tomography is most often used in the 2D variant, which produces cross-sections showing the variability of resistivity along the measuring profile together with its depth. The results of ERT measurements are presented as profiles of apparent resistivity with geophysical interpretation.

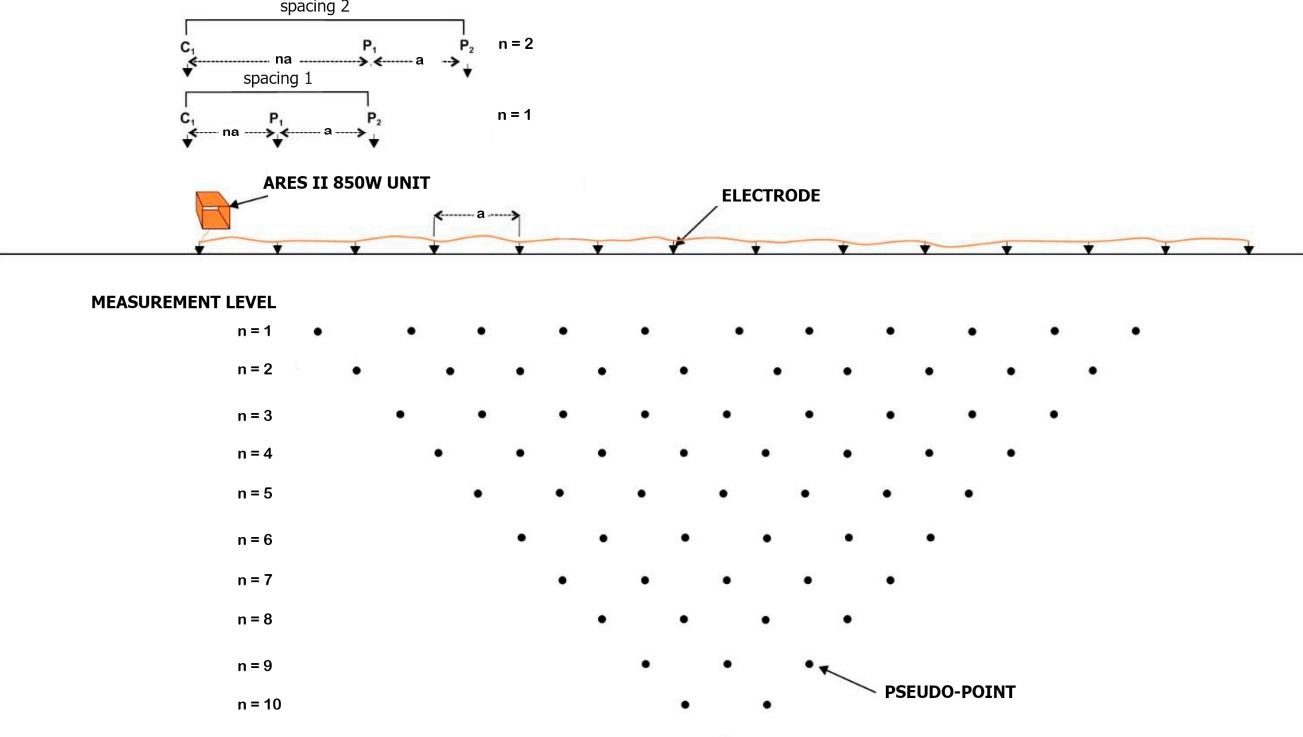

**Figure 6.** Scheme of ERT measurement with a three-electrode system method.

## 4. Results and Discussion

The development of a 3D geospatial database of former mining activity allowed us to map the location and geometry of underground workings with respect to the present-day topography of the study area. The GIS-based geodatabase stores features representing individual vertical, horizontal, and inclined drifts, with information on their depth below the surface, heading, and geometry. The archival documentations, and thus the resulting database, cover the entire area of test field three. These were used to set out test field three and tests sites within its border.

Interpreting the gravimetry results, the areas at the greatest risk of post-mining deformations, related to the consolidation of the ground caused by mining, are associated with the occurrence of negative gravity anomalies resulting from the shortage of mass in the substratum.

The results of microgravimetry data interpretation (Figure 7) show that the Bouguer anomaly has a streaked, alternating pattern visible in the central and northern parts of the test field. From the center of the area to the south, there is a larger, relatively negative anomaly, reaching a minimum of −14.21 mGal at the southern border, near the anthropogenic lake above the excavated coal deposit. There are three streaks of relatively negative

anomalies from the center in the NE part. These negative anomalies are associated with the lighter formations, also of anthropogenic origin, that fill the depressions. The positive anomalies of the SW-NE direction run from the center and continue beyond the northern and north-eastern boundaries of the area. The positive anomalies are probably caused by tertiary formations, with a higher bulk density in relation to the anthropogenic formations marked in the picture of Bouguer anomalies, in the form of relatively negative anomalous forms. The picture of the Bouguer anomaly in, test field three, is shown in Figure 7a. To focus on anomalies associated with shallow sources, separation of gravity Bouguer anomalies into residual components was done in the frequency domain, using the Butterworth high-pass filter. This filter rejects regional components resulting from deeper geological structures.

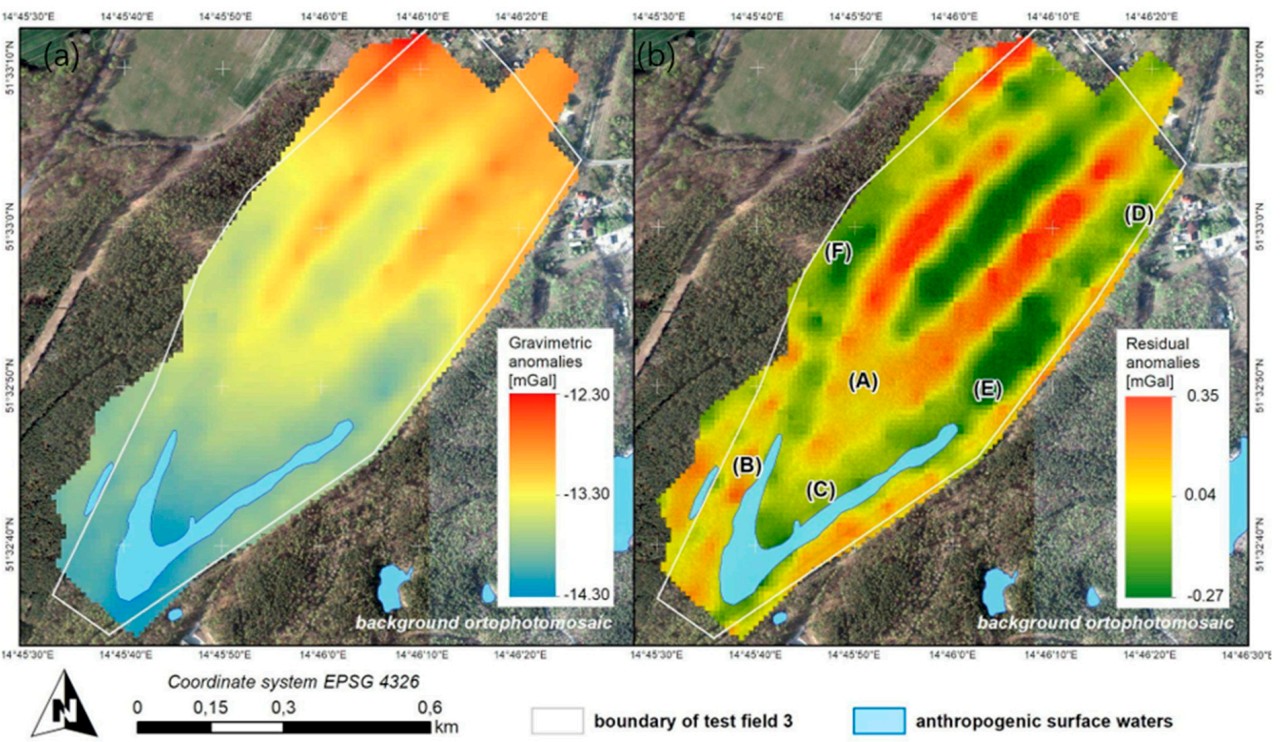

**Figure 7.** Gravimetric map: (**a**) Gravimetric anomalies in the Bouguer reduction; (**b**) Residual anomalies.

The image of residual anomalies is dominated by alternating positive and negative anomalous forms running from the central part of the area towards the north-east (Figure 7b). The two larger forms in the central part merge into one anomaly towards the SW (A). The anomaly near the NE border has three maxima, the largest of which reaches 0.28 mGal. In the southern part, near the anthropogenic lake (B), there are several small positive anomalies with values that do not exceed 0.15 mGal. There are three negative bands between the positive bands with a minimum value of −0.27 mGal (C). The sequences of positive values (in the central and north-eastern part) are contoured by distinct densities separating the areas of positive and negative anomalies. This is probably related to the difference in density between the heavier Miocene formations and lighter anthropogenic formations and brown coal outcrops. In the central and southern parts of test field three, the density contrast is slightly lower, and may indicate transformation of the original environment as a result of the brown coal mining and appearance of anthropogenic deposits (sand, gravel, peat, or silt). With respect to the particular test sites, the one located in the NE part of test field three is covered by a relatively negative residual anomaly, with a minimum of −0.055 mGal (D), located near the south-eastern border of the site. The positive anomalous forms are visible mainly at the edges, and partially in the central part, of this area. This pattern of anomalies is related to the post-mining character of the test site, i.e., the presence of depressions filled with anthropogenic formations of lower density than the surrounding

Miocene forms. In the test site located in the SE part of test field three (E), the distribution of residual anomalies shows an extensive, relatively negative anomaly running from SW to NE. The lowest value of −0.05 mGal is located in the southern part of this site. The image of residual anomalies, as in the case of the previous test site, reflects the post-mining nature of the area, where, next to higher bulk density tertiary deposits, there are lighter anthropogenic formations. The last test site (F) shows the same character.

The 2D and 3D gravimetric modeling methods made it possible to quantify the image of the rock mass in the light of gravimetric data. The Bouguer anomaly maps confirmed the glaciotectonic character of the geological structure, emphasized the geostructural forms, and helped to define the extent of possible land deformations. The areas exposed to the threat of surface deformations are related to the consolidation of the ground disturbed by mining. In gravimetric images, these are associated with the occurrence of negative anomalies, and result from a shortage of mass. The process of tightening the goafs (underground workings) loosens the sediments above, and thus reduces their density in relation to the intact surroundings. These sites were further investigated with the Electrical Resistivity Tomography surveys. The results, in the form of eight apparent resistivity cross-sections beneath the ERT geophysical profiles, and their geophysical interpretation, indicate three distinct lithological complexes, spatially differing in resistivity and physical-mechanical properties (Table 1). The delineation of boundaries between individual complexes is based only on the values of the physical parameter, which is the resistivity of the geological medium. Their spatial extent, in an exemplary profile corresponding to the profile line marked in Figure 5, is shown in Figure 8.

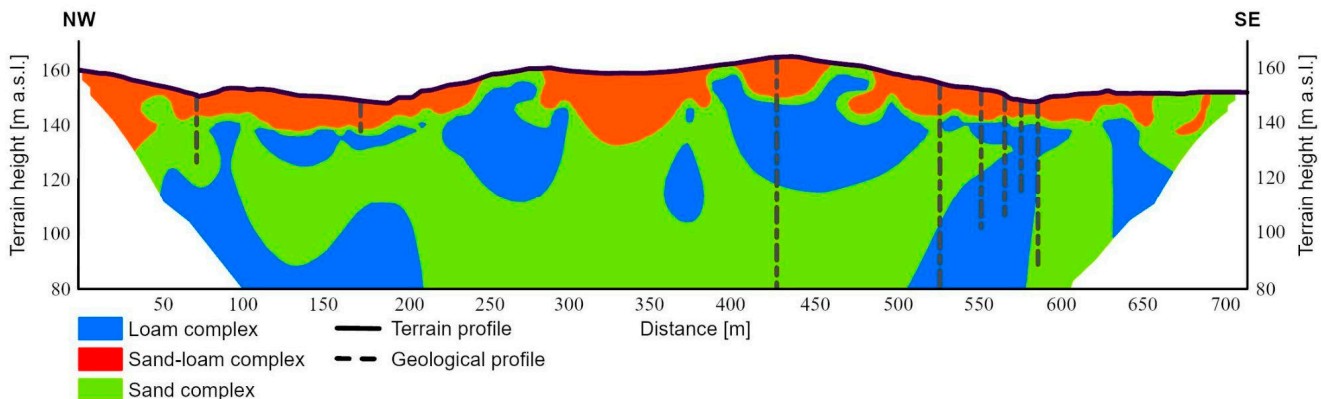

**Figure 8.** Exemplary geoelectrical profile with geophysical interpretation.

**Table 1.** Lithological and physic-mechanical classification of resistance complexes in the study area.

| Complex | Range of Resistivity [Ωm] | Filtration Coefficient [m/s] | Lithological Classification |
|---|---|---|---|
| loam | <40 | Impermeable to poorly permeable ($1 \times 10^{-8}$–$1 \times 10^{-6}$) | Cohesive formations-clay formations (silts, clays, loams) and brown coal |
| sand-loam | 40–200 | poorly permeable to permeable ($1 \times 10^{-6}$–$1 \times 10^{-4}$) | sandy loams (values between 100 and 200 Ωm) indicate predominance of non-cohesive formations (sands, dusty sands, and gravel in the saturation zone). |
| sand | >200 | Permeable ($1 \times 10^{-4}$–$1 \times 10^{-3}$) | Sands and gravels in the aeration zone; anthropogenic formations. |

The ranges of resistivity values of coals versus the typical resistivity values of minerals and rocks (according to [58]).

Based on the adopted criteria, a schematic diagram of the hydrogeological cross-section, with marked extent of deformation zones caused by the exploitation or weathering of brown coal seams (Figure 9), was constructed. The discontinuity zones of poorly permeable and impermeable grounds constitute potential channels for the migration of non-cohesive formations into mining excavations zones. Based on the analysis of these results

in ERT profiles, a map of subsidence risk areas was interpreted, and further geophysical studies provided additional geological information and identified locations of potential ground disturbance of anthropogenic (mining) origin (Figure 10).

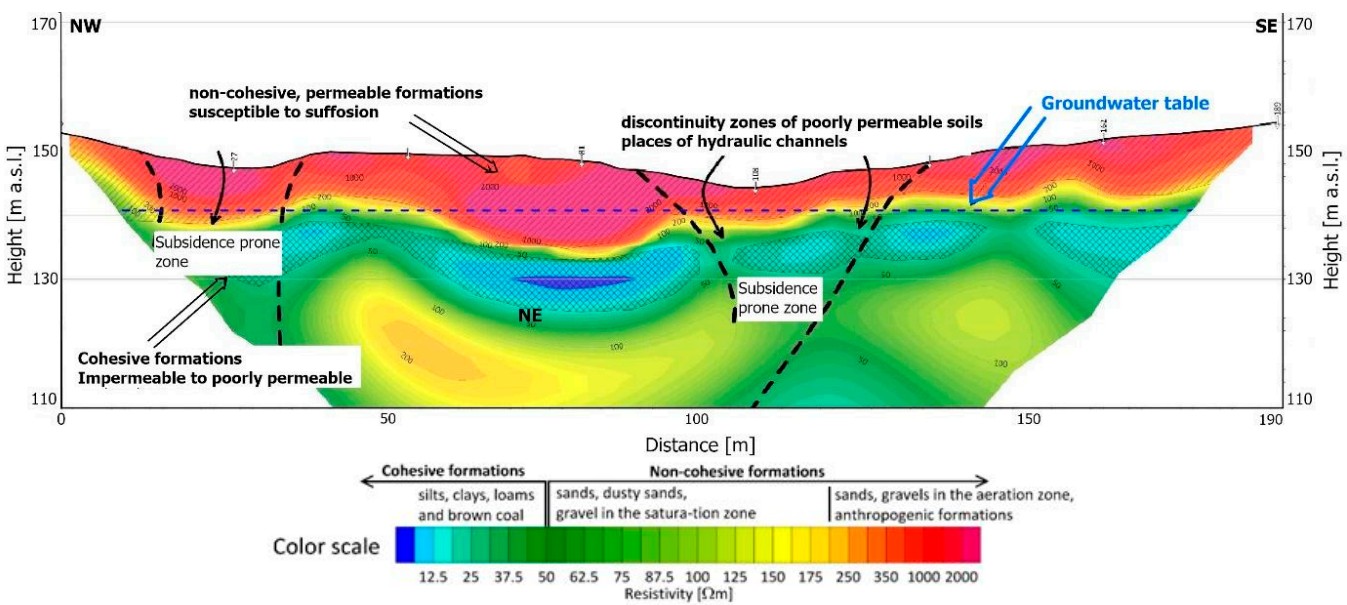

**Figure 9.** Diagram of interpretation of the hydrogeological conditions along profile (shown in Figure 5) in test-field three.

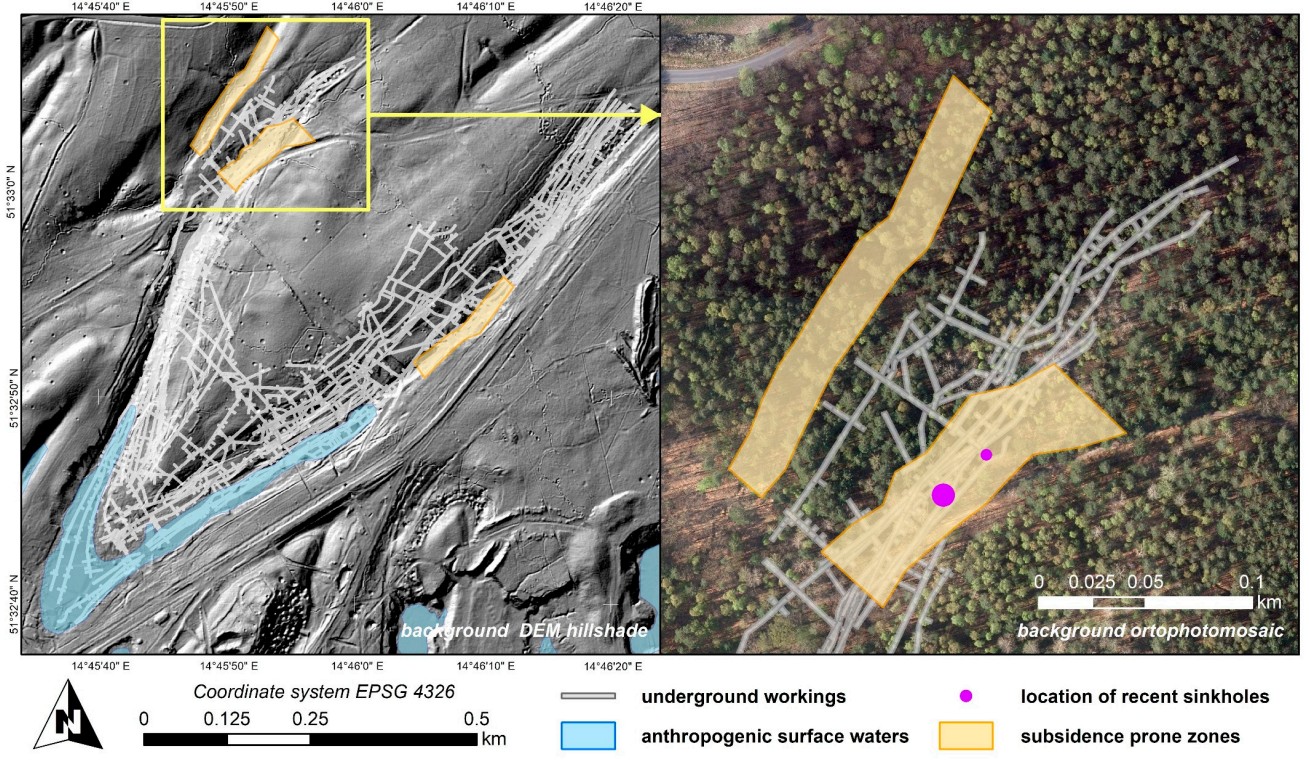

**Figure 10.** Interpreted subsidence risk areas in zones of ground disturbed by mining activity.

The presented results provide information about conditions that can be transferred to other fields of underground mining in the study area, in order to map zones of potential land deformation (sinkhole) occurrence, such as in test field three (Figure 11). The location of this sinkhole is shown in Figures 5 and 10. The feature was spotted in Autumn 2019 by

local forest services, and has been under monitoring with a terrestrial laser scanner since the Spring of 2020. The results of the most recently processed scans (3.2021) indicate an average sinkhole depth of 4 m below surface level.

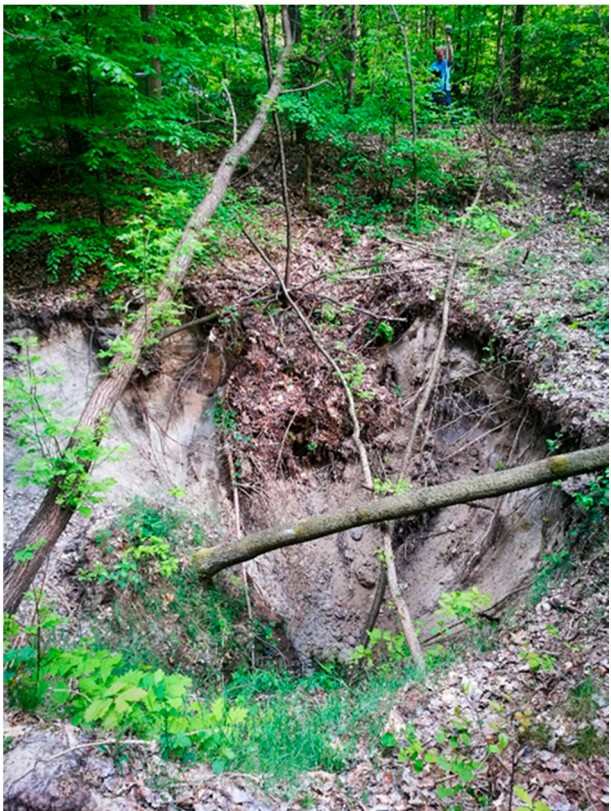

**Figure 11.** Photo of the recent sinkhole in the study area.

This is the first study using geophysical methods in this post-mining area, and probably the first one conducted in a structurally complex glaciotectonic environment. The only known description of mining-related deformations in a comparable area was prepared by [59], and concerns simple inventorying and description of different deformation types, e.g., subsidence basins and sinkholes in former brown coal mines located to the north of the study area. Different types of deformations were also briefly mentioned in [50]. A study of deformations caused by ceased shallow underground brown coal mining, conducted with the application of geophysical and geological surveys in different geological and tectonic settings, was described in [60]. There are numerous studies using different geophysical methods to investigate and map voids resulting from old mining activity, and aimed at identifying potential deformation risk areas. Notable examples are listed in the Introduction section. None of them, however, were conducted in a glaciotectonic area and concerned shallow underground mining of brown coal deposits in complicated conditions of mineral extraction.

## 5. Conclusions

The glaciotectonic Muskau Arch structure was disturbed by long-time mining of brown coal deposits. The effects of past mining in complicated geological conditions, such as secondary deformations, pose a threat to present day development of this area.

This study mapped, for the first time, the old underground workings in relation to present day topography, and identified zones of disturbed ground that are susceptible to the formation of sinkholes.

The results of gravity and ERT geophysical measurements revealed deep anthropogenic transformations of the natural geological conditions, and identified three distinct complexes in the ground.

Furthermore, a model of present-day hydrogeological conditions was proposed in the area of past underground mining. The natural system of separate aquifers there has been transformed, and forms one connected groundwater system.

These results relate the existing sinkholes with shallow underground workings and zones of anthropogenically disturbed ground. The periodic and seasonal fluctuation of the groundwater table causes subsequent watering and dewatering of upper parts of the ground and old mine workings, that activate land deformation processes.

The adopted methodology can be transferred to other parts of underground brown coal mining in the Muskau Arch area, as well as to other regions of shallow mining, and confirms the effectiveness of the adopted approach in determining subsidence risk zones in glaciotectonic areas.

**Author Contributions:** Conceptualization, J.B., E.W. and J.K.; methodology, J.B., E.W. and L.K.; software, J.B., P.T., J.W., M.B. and A.B.; validation, J.B., E.W., M.B. and M.W.; formal analysis, M.W., P.T., E.W. and L.K.; investigation, N.B., D.J. and M.B.; resources, J.K., L.K. and J.W.; data curation, P.T., N.B., A.B. and J.W.; writing—original draft preparation, J.B., E.W. and M.W.; writing—review and editing, A.B., N.B. and P.K.; visualization, J.B., A.B., J.K. and P.T.; supervision, J.B. and E.W.; project administration, J.B. and E.W.; funding acquisition, J.B. All authors have read and agreed to the published version of the manuscript.

**Funding:** This research was financed by the National Science Centre financed OPUS-17 project, entitled "Genesis and course of anthropogenic and natural deformations of the terrain in post-mining areas of the former brown coal mine Babina", no. 2019/33/B/ST10/02975.

**Institutional Review Board Statement:** Not applicable.

**Informed Consent Statement:** Not applicable.

**Acknowledgments:** The authors would like to express their gratitude to the "Lipinki" State Forest District Service for their assistance during the carrying out of the research.

**Conflicts of Interest:** The authors declare no conflict of interest.

## Abbreviations

| | |
|---|---|
| ALOS | Advanced Land Observation Satellite |
| DEM | Digital Elevation Model |
| ERT | Electrical Resistivity Tomography |
| GIS | Geographic Information System |
| GNSS | Global Navigation Satellite Systems |
| GPR | Ground Penetrating Radar |
| InSAR | Interferometric Synthetic Aperture Radar |
| MIS | Marine Oxygen Isotope Stage |
| RI | Resistivity Imaging |

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
