# Peer review of "Geophysical Research of Secondary Deformations in the Post Mining Area of the Glaciotectonic Muskau Arch Geopark—Preliminary Results"

_applsci, doi:10.3390/app12031194_

Round 1

Reviewer 1 Report

Dear authors, I attached my corrections in the PDF.

All the best.

Author Response

Dear Reviewer,

Please find our response in the attached file.

Kind regards

Jan Blachowski

Author Response

(The authors gave the same response as above.)

Reviewer 3 Report

The Abstract: it is not in the usual/required structure, which'd be: (1) what (2) how (3) where (4) why have I made something and (5) what is final conclusion. These all must be mention here and nothing else.

Lines 69-71: move to the Acknowledgements

Introduction: way too many methods are mentioned as 'possible tools', but never applied in the real part of the paper. Shorten this and please focus on the real, more detailed purpose of the presented work (which is clearly only a small part of the project mentioned in Lines 69-71...)

Point 4.3: is it required in such details? Two paragraphs and some references must be enough to show, what ERT is, anything else is required only if it is a significant modification of the textbook method.

Fig 7: however I accept the Bouguer map of part (a), after reading the paper, I’m not convinced that part (b) is a real effect in such fine gravity resolution (tenths-hundredths of milligals) and not a processing artifact. Any reasons needed for the conclusions (real ones I mean), are in the part (a) of the figure.

Sections in Figs 8 & 9: please locate them clearly in Fig 10 (or in Fig 5).

Conclusions: all of my problems with the paper are focused here. No additive information is given here about the study area, which I cannot guess without reading the paper. Does the area indeed have "complicated geometry"? Is THIS a conclusion?? Please, give here more focused statements, even small ones, as „message to the world”, please. The present conclusions chapter contains nothing new.

Author Response

(The authors gave the same response as above.)

Round 2

Reviewer 2 Report

The authors answered my questions and the paper has been well revised.